# Experimental Study on Geopolymerization of Lunar Soil Simulant under Dry Curing and Sealed Curing

**DOI:** 10.3390/ma17061413

**Published:** 2024-03-20

**Authors:** Jinhui Gu, Qinyong Ma

**Affiliations:** 1School of Civil Engineering and Architecture, Anhui University of Science and Technology, Huainan 232001, China; a447781076@163.com; 2Engineering Research Center of Underground Mine Construction, Ministry of Education, Anhui University of Science and Technology, Huainan 232001, China

**Keywords:** lunar soil simulant, geopolymer, NaOH, dry curing, sealed curing, microscopic results

## Abstract

The construction of lunar surface roads is conducive to improving the efficiency of lunar space transportation. The use of lunar in situ resources is the key to the construction of lunar bases. In order to explore the strength development of a simulated lunar soil geopolymer at lunar temperature, geopolymers with different sodium hydroxide (NaOH) contents were prepared by using simulated lunar regolith materials. The temperature of the high-temperature section of the moon was simulated as the curing condition, and the difference in compressive strength between dry curing and sealed curing was studied. The results show that the high-temperature range of lunar temperature from 52.7 °C to 76.3 °C was the suitable curing period for the geopolymers, and the maximum strength of 72 h was 6.31 MPa when the NaOH content was 8% in the sealed-curing mode. The 72 h strength had a maximum value of 6.87 MPa when the NaOH content was 12% under dry curing. Choosing a suitable solution can reduce the consumption of activators required for geopolymers to obtain unit strength, effectively reduce the quality of materials transported from the Earth for lunar infrastructure construction, and save transportation costs. The microscopic results show that the simulated lunar soil generated gel substances and needle-like crystals under the alkali excitation of NaOH, forming a cluster and network structure to improve the compressive strength of the geopolymer.

## 1. Introduction

As the nearest celestial body to the Earth, the unique space environment and abundant mineral resources of the moon make it a key springboard for deep space exploration [1]. Landing on the moon to exploit lunar resources and building a lunar base is the focus of aviation competition among countries. The construction of lunar bases and lunar infrastructures is an important task of lunar exploration projects. The construction of lunar roads is the key to improving the functionality and operational efficiency of lunar bases [2]. The lunar surface is rugged, the gravity is low, the lunar soil is loose, and the adhesion is strong, which makes it easy for a lunar rover to slip, sink, or even become unable to move [3]. It is easy for a lunar rover to lift the lunar surface soil while driving and attach it to the solar panel to make it invalid. In addition, if the loose lunar soil is attached to the positions of the important parts inside the lunar rover, it can easily cause the lunar rover to fail or even short-circuit. The Chinese Yutu lunar rover has a top speed of only 0.2 km/h. Building roads with a certain structural strength on the lunar surface can effectively improve the speed and safety of lunar equipment transportation, thereby further improving the transportation efficiency of goods and providing a basic guarantee for lunar scientific research activities and production and construction. According to NASA‘s report, the cost per kilogram of material sent to the moon‘s orbit is more than USD 20,000 [4]. How to use the moon‘s in situ resources to establish a lunar scientific research base has become a research hotspot in the scientific and engineering communities [5,6].

Alkali-activated lunar regolith is considered to be a promising in situ resource utilization technology for the moon [7,8,9]. It is highly feasible to use lunar regolith as a raw material for geopolymerization to prepare lunar road materials. Studies have shown that geopolymer materials have good physical and mechanical properties [10]. A geopolymer is an inorganic polymer formed by the alkali excitation of silicon–aluminum material, so silicon–aluminum material is the precursor material for the preparation of geopolymers.

According to statistics, the content of silicon–aluminum oxide in the ‘Apollo‘ lunar soil sample is more than 60%, indicating that the lunar soil meets the conditions for preparing geopolymer precursors. The alkaline solution provides hydroxide ions for the polymerization reaction of the simulated lunar soil, which promotes the disintegration of the silicon–aluminum structure in the simulated lunar soil [11]. The lunar regolith itself has low strength and does not contain cementitious minerals, but it can be converted into cementitious materials through an alkali excitation reaction. After four reaction stages of deconstruction, reconstruction, condensation, and crystallization [12], it is finally solidified into a geopolymer with a certain structural strength. Sodium hydroxide (NaOH) solution is a commonly used alkali activator, and its content directly affects the performance of the geopolymer.

Grugel et al. [13] prepared sulfur-rich geopolymers using JSC-1 simulated lunar soil. Although the problem of the material source was solved, it deforms and melts when the temperature reaches 119 °C, thereby reducing its strength. Alexiadis et al. [14] used a JSC LUNAR-1A lunar soil simulant to prepare a geopolymer with NaOH solution as an alkali activator and found that the compressive strength of the lunar soil simulant geopolymer was significantly different according to the concentration of the sodium hydroxide solution. Mills et al. [15] compared the mechanical properties of alkali-activated lunar-soil-like geopolymers prepared in vacuum extreme-high-temperature (600 °C) and low-temperature (−80 °C) environments and found that the high-temperature environment was more conducive to the curing of geopolymers. Xiong et al. [16] treated a simulated lunar soil geopolymer excited with a NaOH solution with high- and low-temperature cycles of 120 °C and −30 °C and found that the compressive strength of the geopolymer was even improved after 20 cycles. Pilehvar et al. [17] used a NaOH solution to prepare a DNA-1 lunar soil simulant geopolymer with a uniaxial compressive strength of 16 MPa. Therefore, it is scientific and promising to use simulated lunar soil as the raw material and sodium hydroxide solution as the activator to prepare a simulated lunar soil geopolymer for the study of lunar road construction materials based on the two aspects of material selection and method use.

Most researchers choose to maintain samples in a constant-temperature and -humidity environment, but the temperature of the lunar surface changes with time and is very dry. When the temperature of the lunar surface is 50–80 °C, it meets the polymerization conditions for a simulated lunar soil geopolymer. Although the water inside the geopolymer will be lost in a dry environment, it can be collected and reused, which is an important part of saving the transportation costs of lunar resources. Weng et al. [18] found that water played a role as a medium in the process of geopolymerization. Water only existed as a reactant in the dissolution stage and existed in the form of products in the other stages. Wang et al. [19] prepared a condensate water circulation system and cured the simulated lunar soil geopolymer at the lunar ambient temperature. The results showed that 98% of the water could be recycled during the curing process of the simulated lunar soil geopolymer. At present, there are few reports on the comparison of the mechanical properties of simulated lunar soil geopolymers in a water loss curing environment and water retention curing environment. Therefore, a simulated lunar soil geopolymer was prepared by using the in situ temperature of the moon, and the mechanical properties of the simulated lunar soil geopolymer under dry curing (a water loss environment) and sealed curing (a water retention environment) were studied. The difference and the influence of a dry-curing environment on the simulated lunar soil geopolymer can provide some reference for the construction of lunar roads.

In this experiment, five high-temperature periods were selected according to the temperature change curve of the moon at 45° latitude near the landing area of Chang ‘E-5. Three kinds of alkali-activated lunar soil simulants with different NaOH concentrations were prepared using volcanic ash lunar soil simulants as the raw materials, and the lunar soil simulants were cured by dry curing and sealed curing. The mass loss rate and side length shrinkage rate of the geopolymer in a dry environment at the lunar surface temperature were studied. By comparing the differences in the compressive strength and mass strength efficiency of the geopolymer under different schemes, the optimum curing temperature scheme and NaOH concentration of the simulated lunar soil material were determined, which provides an experimental basis for the application of simulated lunar soil geopolymers in lunar pavement construction.

## 2. Materials and Methods

### 2.1. Materials

The raw material used in the experiment was basaltic volcanic slag from Lingshou County, Hebei Province. The simulated lunar soil materials for preparing the geopolymers were obtained by screening and grinding them, as shown in Figure 1.

The XRD pattern of the experimental lunar soil simulant is shown in Figure 2. It can be seen that the main phases of the lunar soil simulant were albite, anorthite, andesine, augite, and olivine. This is consistent with the main mineral composition in real lunar soil [20].

Table 1 lists the chemical composition of the real lunar soil collected from the Apollo probe and the simulated lunar soil materials. In order to make the particle size of the simulated lunar soil material closer to that of the real lunar soil, the fine simulated lunar soil material was screened to obtain the particle size distribution, as shown in Figure 3 [21].

The sodium hydroxide (NaOH) solution was prepared by mixing sodium hydroxide particles with distilled water as the mixing water and alkali activator. The purity of the NaOH used was 96% (mass fraction).

### 2.2. Curing Section

Figure 4 shows the real ambient temperature of the lunar surface at different latitudes [22]. This experiment was intended to study the mechanical properties of the simulated lunar regolith geopolymer at the real temperature of the moon, so the temperature change line at 45° latitude closest to the landing point of Chang ‘E-5 was selected. According to the method in [23], since the duration of a day on the moon is equal to 30 days on the Earth, the time on the moon must be converted into the time on the Earth. The linear interpolation method was used to determine the temperature value every 12 h, and the temperature section above 0 °C was intercepted. Marks A, B, C, D, and E correspond to 222 h–294 h, 270 h–342 h, 318 h–390 h, 366 h–438 h, and 414 h–486 h, as shown in Figure 5. Table 2 lists the curing temperature schemes to simulate the temperature change on the lunar surface at 45 ° latitude.

### 2.3. Geopolymer Synthesis

The solid NaOH was dissolved in distilled water to prepare the NaOH solution. After cooling to room temperature, it was mixed with simulated lunar soil powder to prepare simulated lunar soil geopolymer pastes. The coordination of the lunar soil simulant geopolymer is shown in Table 3. The mixed simulated lunar soil geopolymer pastes were fully mixed in the mortar mixer. After the mixing was completed, the pastes were poured into molds of 50 mm × 50 mm × 50 mm and then vibrated to remove air bubbles in the pastes, and a polyethylene film was wrapped and then placed in an electric drying oven (101-00B, Shangcheng Instrument Manufacturing Co. Ltd., Shaoxing, China) according to the A–E schemes. After 24 h of curing, the molds were removed, one group of samples was stopped curing and their 24 h compressive strength was immediately measured, one group of samples was sealed with a PE-sealed bag and put into the electric drying oven for sealed curing, and the other group of samples without sealing treatment was directly put into the electric drying oven for dry curing to simulate the dry environment on the moon. The curing methods and steps are shown in Figure 6.

### 2.4. Test Method

A uniaxial compression test machine (DNS-300, Changchun Machinery Institute, Changchun, China) was used to measure the compressive strength of the samples at 24 h and 72 h under each curing scheme. The loading rate was 0.5 mm/min, and the obtained compressive strength was the average strength of the three samples.

An electronic balance was used to weigh the quality of the simulated lunar soil geopolymer specimen in the B–E temperature schemes with the dry-curing method. The accuracy of the electronic balance was 0.01 g, and the final mass measured was the average of the mass of the three specimens.

A vernier caliper was used to measure the axial length of the simulated lunar soil geopolymer specimen in the B–E temperature schemes under dry curing. The accuracy of the vernier caliper was 0.01 mm, and the measured final axial length was the average of the axial lengths of the three specimens.

The internal microstructures of the geopolymers in the curing section with different NaOH contents at 72 h were characterized by scanning electron microscopy (SEM, FlexSEM1000, Japan).

## 3. Results

### 3.1. Compressive Strength

The compressive strengths of the geopolymer at 24 h and 72 h under the same curing scheme are shown in Figure 7. The curing temperature within 24 h of scheme A was low, which delayed the bonding and hardening of the geopolymer. Therefore, the 24 h compressive strength of each group of specimens under this scheme was zero, and the compressive strength was measured only after sealed curing for 72 h. When the content of NaOH was 8%, the sample was damaged and the bearing capacity was lost when the sample was dry-cured for 72 h in scheme C, as shown in Figure 8. This was due to the rapid evaporation of water in the geopolymer, whereby the specimen shrank, and micro-cracks were first generated outside. As the shrinkage intensified, the cracks extended inward until the cracks ran through the entire specimen to break the specimen and it lost its compressive capacity.

In Figure 7a, it can be seen that compared with the compressive strength of the five temperature curing schemes at the same NaOH concentration for 24 h, the compressive strength in the C and D schemes was significantly higher than that in the other groups. When the NaOH concentration was 8%, the strength was the largest, 3.24 MPa and 3.25 MPa, respectively. The compressive strength of the specimens in groups C and D at the same NaOH concentration was comparable because the curing temperature of the two groups for 24 h was about 75 °C, and was higher than in other schemes. It can be seen that in the high-temperature curing environment, it was beneficial to accelerate the combination reaction of the NaOH solution and lunar soil simulant material, thereby improving the strength of the geopolymer. On the contrary, the lower initial curing temperature hindered the dissolution and reaggregation of silicates and aluminates, making the geopolymer’s compressive strength relatively low [24]. Therefore, the compressive strength of the geopolymers in schemes B and E for 24 h was small, while in scheme A, the strength of the specimen was not formed due to the low temperature.

In Figure 7b,c, it can be seen that when cured for 72 h, except for the geopolymer with an 8% NaOH content in group C, which lost its strength due to shrinkage and rupture, the compressive strength in the other groups was higher than that at 24 h. At 72 h, the geopolymer with a 12% NaOH content under dry curing in scheme D obtained the maximum compressive strength of 6.87 MPa, which was 114.7% higher than that at 24 h. The geopolymer with an 8% NaOH content in scheme D obtained the maximum strength of 6.31 MPa after 72 h under sealed curing, which was 94.2% higher than that at 24 h. When the content of NaOH was 4%, the compressive strength in group E after 72 h of sealed curing was 0.3 MPa, which was only 44.1% of the compressive strength at 24 h, and the strength showed negative growth. This may be due to the low content of NaOH, on the one hand, and, on the other hand, due to the decrease in temperature, the dissolved silicate and aluminate could not agglomerate well, which hindered the process of the hydration reaction.

The 72 h strength of the specimen with a 12% NaOH content in scheme D under sealed curing increased by only 7.2% compared with the 24 h strength, which was much smaller than the strength growth under dry curing. Because the curing temperature in this scheme was gradually reduced from 76.3 °C to 52.7 °C, the decrease in temperature led to a decrease in the rate of polymerization reaction. In the process of dry curing, although the temperature was also reduced, the water in the geopolymer also constantly evaporated, which made the concentration of NaOH increase accordingly, which was conducive to the development of the strength of the geopolymer. The strength of the latter was 100.3% higher than that of the former.

In order to show the development and change in strength more clearly, Figure 7b also shows the ratios of the 24 h compressive strength to the 72 h final strength of the geopolymer with an 8% NaOH concentration in the sealed-curing mode, which were 0, 47.3%, 60.6%, 51.5%, and 85.9%, respectively. Figure 7c shows the ratios of the 24 h compressive strength to the 72 h final strength of the geopolymer with a 12% NaOH concentration in the dry-curing mode, which were 36.6%, 63.3%, 46.6%, and 63.4%, respectively. This shows that the early strength development of the geopolymer was sensitive to temperature, and the subsequent temperature change affected the strength growth of the geopolymer.

### 3.2. Mass Loss Rate

For the specimens under dry curing, the water was lost with the increase in the curing time, which manifested as a decrease in specimen quality. An electronic balance was used to measure the mass of the simulated lunar soil geopolymer specimen when it was cured for time t (t = 24 h, 36 h, 48 h, 60 h, and 72 h), and the mass loss rate was calculated to measure the water loss in the specimen. The formula was:(1)$$q_{m}=\frac{m_{t}-m_{24h}}{m_{24h}}\times 100\%$$

In the formula, *q*_m_ represents the mass loss rate of the simulated lunar soil geopolymer specimen; *m*_t_ denotes the mass of the specimen at time t; and *m*_24h_ represents the initial mass of the specimen when it was cured for 24 h.

The mass loss rate of the geopolymer under the B–E temperature schemes in the dry environment is shown in Figure 9. When the curing time was 72 h, the geopolymer with a NaOH content of 4% had higher mass loss rates under the B–E temperature schemes, which were 22.89%, 21.05%, 21.43%, and 13.91%, respectively. The mass loss rate in scheme E was much smaller than that in schemes B, C, and D, which was due to the lower curing temperature and the lower water loss rate. When the NaOH content of the local polymer was 8%, the mass loss rate steadily increased under the B and C curing schemes, and the moisture in the geopolymer continued to evaporate in the high-temperature dry environment. The mass loss rate was small in the D scheme, which was inevitably related to the subsequent temperature reduction. When the NaOH content in the local polymer was 12%, the mass loss rate during the curing for 72 h in each curing scheme was small. The final mass loss rate in the D scheme was 11.64%, while the final mass loss rate in the E curing scheme was only 4.92%. This shows that 12% NaOH has a good water retention effect on geopolymers in a dry environment.

### 3.3. Shrinkage Rate

A simulated lunar soil geopolymer and volcanic ash geopolymer are essentially the same, and they will produce shrinkage deformation during the hardening process [25,26]. This shrinkage mainly includes chemical shrinkage caused by the alkali-activated hydration reaction, temperature shrinkage caused by the temperature change, and drying shrinkage caused by water loss.

During the test time, there were no cracks or shrinkage damage in the simulated lunar regolith geopolymer specimens in the sealed-curing environment, while shrinkage cracks appeared in some specimens in the dry-curing environment. Therefore, a vernier caliper was used to measure the axial length of the dry-cured simulated lunar soil geopolymer specimen when it was cured for time t (t = 24 h, 36 h, 48 h, 60 h, and 72 h). The shrinkage rate was calculated, and the shrinkage deformation of the specimen was analyzed. The formula was:(2)$$s_{l}=\frac{l_{t}-l_{24h}}{l_{24h}}\times 100\%$$

In the formula, *s*_l_ represents the shrinkage rate of the simulated lunar soil geopolymer specimen; *l*_t_ represents the axial length of the specimen at time t; and *l*_24h_ represents the initial axial length of the specimen when it was cured for 24 h.

Figure 10 shows the shrinkage change trend in the simulated lunar soil geopolymer with different sodium hydroxide contents under different temperature curing schemes with the increase in the curing time in the dry-curing environment. When the NaOH content in the simulated lunar soil geopolymer was constant, the shrinkage rate in the different temperature curing schemes was different. It can be seen in the diagram that when the content of NaOH was 4%, the shrinkage of the geopolymer cured for 72 h in the B and E schemes was 3.49% and 3.16%, respectively, which was much higher than the values of 0.89% and 1.17% in the C and D schemes, and the B and E schemes had high shrinkage rates of 3.13% and 2.80% at 12 h after mold removal. The geopolymer with an 8% NaOH content had a shrinkage rate of 2.77% when it was cured for 72 h in scheme B, and the shrinkage rate steadily increased with the increase in the curing time. The shrinkage rate in scheme C was also at a high level, reaching 2.32% at 60 h. However, due to the high dry-curing temperature and the rapid evaporation of water, the geopolymer contracted and ruptured and lost its bearing capacity. When the NaOH content in the local polymer was 12%, the shrinkage rate of the specimen was at a low level in each scheme, and the mass shrinkage was small, indicating that the shrinkage effect caused by water loss was small.

Under the four curing schemes of B–E, the three geopolymers with different NaOH contents showed that the shrinkage rate increased with the increase in the curing time. This is because in the dry environment, the water in the geopolymer continuously evaporated, and the gel phase shrank accordingly. When the curing time was 72 h, the shrinkage rates of the three NaOH-containing geopolymers under the B curing scheme were the largest. This may have been due to the gradual increase in the curing temperature under the B scheme, which was more conducive to the formation of the gel and the evaporation of water. The three-phase effect of chemical shrinkage, drying shrinkage, and temperature shrinkage increased, resulting in an increase in the shrinkage rate of the geopolymers. After the E curing scheme for 48 h, the growth rate of the shrinkage rate of the geopolymers with the three NaOH contents was significantly reduced. The shrinkage rate of the geopolymers with a NaOH content of 4% remained almost unchanged with time, while the shrinkage rate of the geopolymers with NaOH contents of 8% and 12% even decreased. This is because the E scheme had a lower curing temperature after 48 h, and the chemical shrinkage, drying shrinkage, and temperature shrinkage were significantly reduced, which eventually led to a slight expansion of the geopolymer.

### 3.4. Mass Strength Efficiency

In the construction of a lunar base, considering the transportation cost of materials, the mass of materials transported from the Earth should be minimized, and *c*_m_ is defined as the mass strength efficiency (MSE) to measure the mass-saving degree of NaOH and water. The formula is:(3)$$c_{m}={\left[\frac{1}{f}\times \frac{m_{24h}}{m_{T}}\times \frac{\left(m_{\mathrm{NaOH}}+m_{w}\right)\times 100}{m_{\mathrm{lss}}}\right]}^{-1}$$

In the formula, *c*_m_ represents the reciprocal of the mass ratio of the sum of NAOH and water forming a 1 MPa strength to the mass ratio of the lunar soil simulant. The larger the *c*_m_, the smaller the total mass of NaOH and water required to form the unit strength of the geopolymer; thus, less material mass is carried from the Earth. *m*_24h_ represents the quality of the geopolymer when it was cured for 24 h; *m*_T_ represents the mass of the geopolymer cured for time T, where T = 24 h or 72 h; *m*_NaOH_ represents the mass of NAOH; *m*_w_ represents the mass of water; and *m*_lss_ represents the mass of the lunar soil simulant. *f* represents the compressive strength value. The mass strength efficiency of the simulated lunar regolith geopolymer at 24 h and 72 h under each test scheme is shown in Figure 11.

It can be seen in Figure 11a that the *c*_m_ value at 24 h was *c*_m_ (8%) ≥ *c*_m_ (12%) ≥ *c*_m_ (4%) in each temperature curing scheme, indicating that too little and too much NaOH reduces the mass strength efficiency of lunar soil simulant materials within 24 h. Figure 11b shows that under the condition of 72 h dry curing, the *c*_m_ of the geopolymer with a 12% NaOH content had a high level in each curing scheme group. In the D curing scheme, the maximum compressive strength was reached at 72 h, and the *c*_m_ reached a maximum value of 1.85 MPa, which was 143% higher than that at 24 h. It can be seen in Figure 11c that the geopolymer with an 8% NaOH content had a higher *c*_m_ value in the 72 h sealed-curing environment. Similarly, in the D scheme, the *c*_m_ had a maximum value of 1.66 MPa, an increase of 93% over 24 h, and the compressive strength also reached the maximum value.

It can be seen that 8% and 12% NaOH had a significant effect on the strength growth of the geopolymers, while the 24 h *c*_m_ values of the geopolymers with a 4% NaOH content under each curing scheme were smaller than those of the former two, and the *c*_m_ value at 72 h was smaller than the 24 h *c*_m_ value. The growth was even negative based on the *c*_m_ value after curing for 72 h in the E scheme, indicating that 4% NaOH had a weak effect on the strength growth of the geopolymers.

### 3.5. Stress–Strain Curve

Since the simulated lunar soil geopolymers had similar stress–strain curves, the 72 h compressive stress–strain curve of the geopolymer under the D scheme was selected as a typical representative, as shown in Figure 12.

It can be seen in Figure 11 that the NaOH content had a significant effect on the stress–strain curve of the simulated lunar regolith geopolymer. When the content of NaOH was 4%, the compressive strength of the geopolymer was small, the peak strain was small, the stress–strain curve was flat, and the ductility and toughness of the geopolymer were good. The geopolymer under 12% NaOH sealed curing and 8% NaOH dry curing had a higher peak stress, the stress–strain curve was of the “thin and narrow“ type, and the compressive strength was higher.

Although the effects of different NaOH contents and different curing conditions on the stress–strain curve shapes of the geopolymers were different, they all showed a consistent brittle failure mode. As shown in Figure 13, The failure process was approximately divided into the following four stages: the compaction stage (I), the linear elastic stage (II), the yield deformation stage (III), and the failure stage (IV). The specific performance was as follows: in stage I, the internal pores of the specimen were compressed under loading, the specimen was gradually compacted, the stress increased in the “concave“ form, and the specimen was not obviously damaged. In stage II, the internal pores of the specimen could no longer be compressed, the specimen was in the stage of linear elastic deformation, the stress increased almost in a straight line, and micro-cracks appeared on the surface of the specimen. In stage III, the stress gradually increased in the “convex“ form until it reached the peak point, and the crack width of the sample gradually expanded and slowly extended to both ends. In the IV stage, the specimen was unable to continue to bear the load, the stress–strain curve decreased rapidly, and the specimen exhibited brittle failure and penetrating cracks.

### 3.6. Energy Characteristics

According to the first law of thermodynamics and the law of conservation of energy, the input energy of the press loaded on the simulated lunar soil geopolymer during the static compression test was:*U* = *U*_e_ + *U*_d_(4)

In the formula, *U* is the input energy of the press to the simulated lunar soil geopolymer; *U*_e_ is the elastic energy of the lunar soil simulant; and *U*_d_ is the dissipation energy, in MJ/m^3^.

The calculation formulas of *U*, *U*_e_, and *U*_d_ in the compression process of the simulated lunar soil geopolymer samples are as follows: (5)$$U=\int_{0}^{\varepsilon_{0}}\sigma_{0}Dd\varepsilon_{0}$$
(6)$$U_{e}=\frac{1}{2}\sigma_{0}\varepsilon_{0}\approx \frac{{\sigma_{0}}^{2}}{2E}$$
(7)$$U_{d}=U-U_{e}$$

In the formulae, *σ*_0_ and *ε*_0_ are the peak stress and peak strain of the simulated lunar soil geopolymer, respectively, and *E* is the slope of stage II in the stress–strain curve.

The relationship curves between the sodium hydroxide content and energy parameters of the group D samples under the two curing methods are shown in Figure 14.

Figure 14a shows that in the dry-curing mode, when the NaOH content was 12%, the elastic energy was the highest and amounted to 0.0423 MJ/m^3^. The ratio of the elastic energy to the input energy was the largest, which was 0.69. The ratio of the dissipation energy to the input energy was the smallest, which was 0.31. It can be seen in Figure 14b that in the sealed-curing mode, when the NaOH content was 8%, the input energy and elastic energy were the largest, which were 0.0612 MJ/m^3^ and 0.034 MJ/m^3^, respectively. The ratio of the elastic energy to the input energy was the largest, which was 0.56. The ratio of the dissipated energy to the input energy was the smallest, which was 0.44. This was similar to the strength characteristics of the simulated lunar regolith geopolymer, so it is speculated that the energy accumulation and dissipation of the simulated lunar regolith geopolymer are related to the compressive strength.

### 3.7. Micro Results and Analysis

Figure 15 shows the scanning electron microscope images of the geopolymers with NaOH contents of 4%, 8%, and 12% after sealed curing for 72 h and a NaOH content of 8% after dry curing for 72 h under scheme D.

It can be seen in Figure 15a that the surface of the geopolymer was covered with agglomerated gel particles, but it showed a relatively loose state. There were large pores in the gel material produced by hydration, and there were unreacted lunar soil simulants. The bulk material indicates that when the NaOH content was 4%, the geopolymer hydration reaction was incomplete, which affected the development of strength.

In Figure 15b, it can be seen that when the NaOH content was 8%, a large number of rod-like crystals were formed on the surface of the geopolymer, and loose gel material groups rarely existed. The rod-like crystals grew in a crisscross manner and formed a dense network structure, which greatly improved the strength of the geopolymer and had a high degree of polymerization, which was consistent with the results of the compressive strength.

It can be clearly seen in Figure 15c that the hydration products generated when the NaOH content was 12% were uneven. There was a large number of gel substances on the surface of the geopolymer, and there were also holes, indicating that excessive NaOH hinders the hydration reaction of the geopolymer, which is not conducive to the development of strength. In addition, prismatic crystals were also found in the image. He [27] believed that the prismatic crystals produced by the hydration reaction of geopolymers were a special form of sodium silicate impurity, which had an adverse effect on the strength of the geopolymers. This also further explained the reason for the decrease in the compressive strength of the geopolymers when the NaOH content was 12%.

As shown in Figure 15d, the hydration reaction of the geopolymer with a NaOH content of 8% under dry-curing conditions was relatively uniform, the needle-like crystals formed on the surface were distributed in a network, and there were obvious gel particle clusters. This was due to the internal water loss in the geopolymer. The hydration was not complete, and the cementitious material could not further react to form a denser structure, which limited the development of compressive strength. However, at the same time, the dense structure hindered the loss of water outward, which also explains the reason why the mass loss rate of the geopolymer with a NaOH content of 8% was low under this curing scheme. Therefore, a proper amount of NaOH and sufficient water are the key factors for the development of geopolymer strength.

## 4. Discussion

Considering the mechanical properties and microscopic characteristics of the simulated lunar soil geopolymer, two mechanisms can be proposed to clarify its curing under sealed curing and dry curing.

In the sealed-curing environment, the geological polymerization reaction was the main curing method for the simulated lunar soil geopolymer. In this experiment, the geopolymer reaction was mainly related to the curing temperature and the concentration of the NaOH solution. Only when the temperature was high enough, the simulated lunar soil geopolymer could produce sufficient strength in a limited time. Taking the simulated lunar soil geopolymer specimen with a NaOH content of 4% as an example, when the curing temperature during 24 h was 5.9 °C–21.8 °C, the geopolymer had not yet completed the initial curing, and the strength was 0; when the curing temperature during 24 h was 52.7 °C–60.2 °C, the compressive strength of the simulated lunar soil geopolymer specimen was 0.37 MPa; and when the curing temperature during 24 h was 74.9 °C–76.2 °C, the compressive strength of the specimen reached 1.45 MPa. This means that the specimen had a high degree of polymerization reaction in the temperature range of 60.2 °C–76.2 °C. This threshold phenomenon was also found in the tests by Geng [7].

In addition, the solidification of the simulated lunar soil geopolymer was affected by the concentration of the NaOH solution. When the concentration of the NaOH solution was low, the aluminosilicate could not be completely dissolved, and the simulated lunar soil particles could not be fully geologically polymerized. The product was relatively simple, showing a lower compressive strength, which can be observed in Figure 15a. When the concentration of the NaOH solution was too high, the initial product was wrapped around the simulated lunar regolith particles, which can be observed in Figure 15c. Therefore, the early compressive strength of the simulated lunar regolith geopolymer was reduced, and the later geological polymerization rate was slowed down. This rule has also been confirmed in the study by Dai [28].

In the dry-curing environment, the simulated lunar soil geopolymer dehydrated, and its curing mechanism was combined in two ways: one was the chemical curing method of geological polymerization, and the other was the physical curing method of dry silicate combination. According to the recognized geological polymerization mechanism, water itself does not participate in the reaction in the process of geopolymerization but exists as a medium [29]. In the dry-curing environment, when the specimen was in the initial stage of curing, due to the sufficient water in the geopolymer, the curing mode was dominated by geological polymerization. With the continuous evaporation of water, the reaction medium was slowly deprived. The combination of silicate solutes in the interstitial fluid of the lunar soil simulant particles dominated the curing of the geopolymer. Under the same water–binder ratio, NaOH with a higher concentration can introduce more silicates into the interstitial fluid and bond the lunar soil simulant particles together after dehydration, which is similar to the bonding mode of alkali silicate-bonded sand [30].

It is worth noting that although the test found that when the curing temperature gradient was reduced from 76.3 °C to 52.7 °C, the simulated lunar soil geopolymer obtained a better compressive strength under both sealed curing and dry curing, it should also be noted that in a continuous high-temperature dry-curing environment, the specimen may shrink and rupture, so it is also necessary to monitor the physical and mechanical properties of the specimen in a long-term dry-curing environment.

## 5. Conclusions

By using the original temperature of the moon to prepare the simulated lunar soil geopolymer, and simulating the compressive strength of the geopolymer cured in the dry and humid environment of the moon, the following conclusions were drawn:In the dry environment, the geopolymer produced mass loss and shrinkage deformation due to water loss. In the high-temperature environment, the geopolymer with a 4% NaOH content had the fastest mass loss rate. When the NaOH content was 8%, dry cracking shrinkage damage easily occurred; when the NaOH content was 12%, the geopolymer had a better water retention capacity and lower shrinkage deformation.Considering the compressive strength of the geopolymer under the five temperature curing schemes, the D scheme with a moon time of 366 h–438 h and temperature change of 76.3 °C–52.7 °C was the appropriate curing time. When the NaOH content of the local polymer was 8%, the strength reached 6.31 MPa after sealed curing for 72 h. When the NaOH content was 12%, the strength reached 6.87 MPa after dry curing for 72 h.The dry-curing and sealed-curing methods had the maximum mass strength efficiency *c*_m_ under the D temperature scheme. The simulated lunar soil geopolymer *c*_m_ with a NaOH content of 12% reached the maximum value of 1.85 MPa under dry curing, and the *c*_m_ reached the maximum value of 1.66 MPa when the NaOH content was 8% under sealed curing.The microscopic results show that the appropriate amount of NOH made the simulated lunar soil geopolymer generate needle-like crystals and dense structures under the action of alkali excitation, thereby improving the compressive strength of the geopolymer.

## Figures and Tables

**Figure 1 materials-17-01413-f001:**
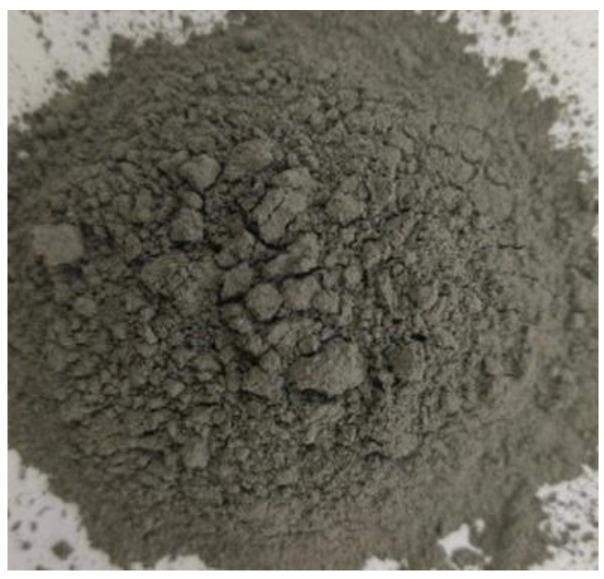
Photograph of experimental lunar soil simulant.

**Figure 2 materials-17-01413-f002:**
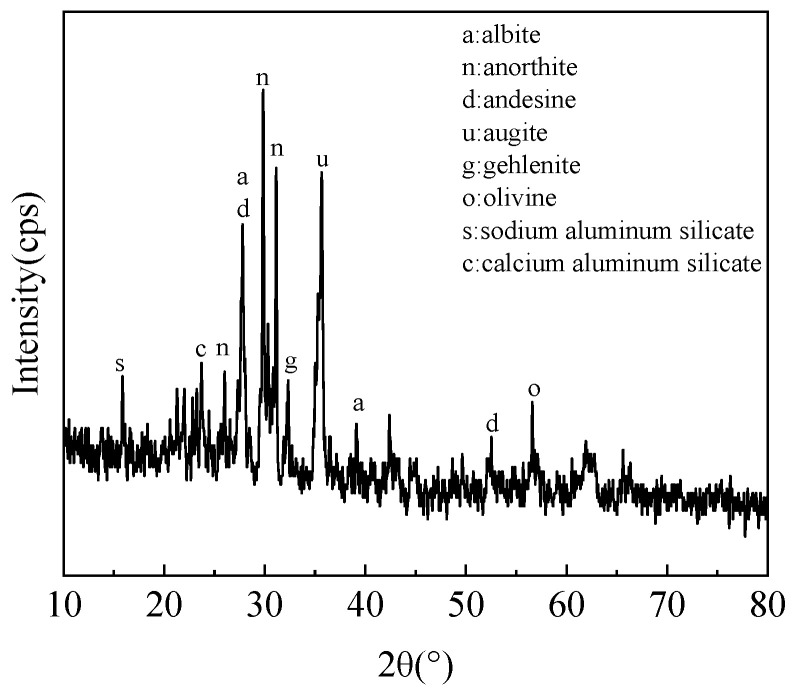
XRD pattern of experimental lunar soil simulant.

**Figure 3 materials-17-01413-f003:**
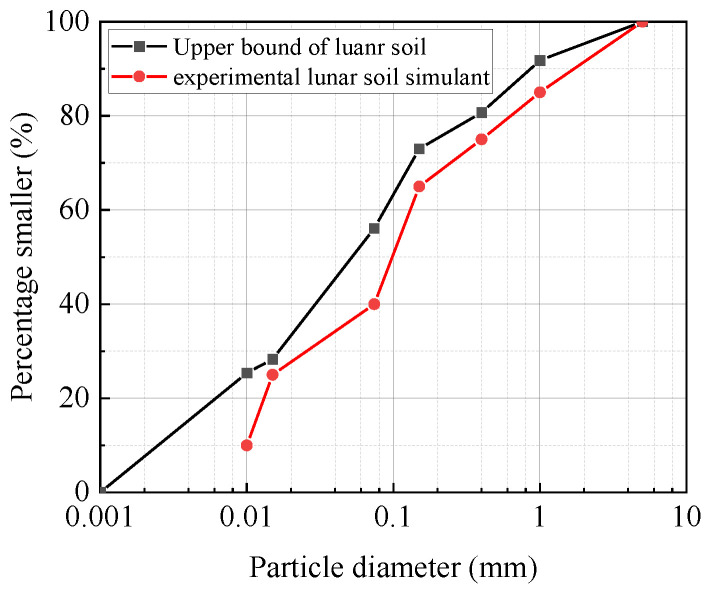
Particle size distribution of experimental lunar soil simulant.

**Figure 4 materials-17-01413-f004:**
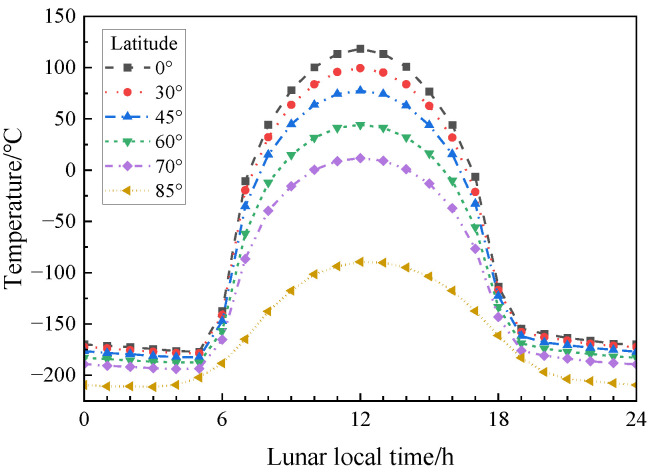
Lunar surface temperature [22].

**Figure 5 materials-17-01413-f005:**
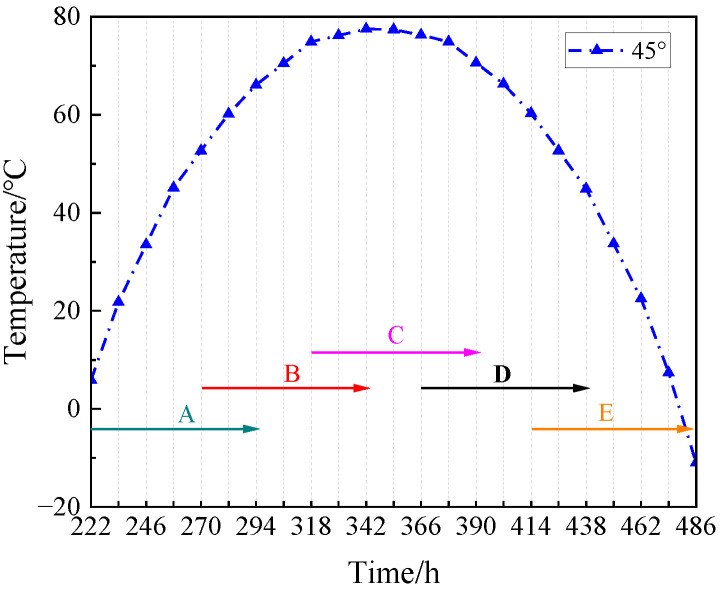
Temperature section from A to E.

**Figure 6 materials-17-01413-f006:**
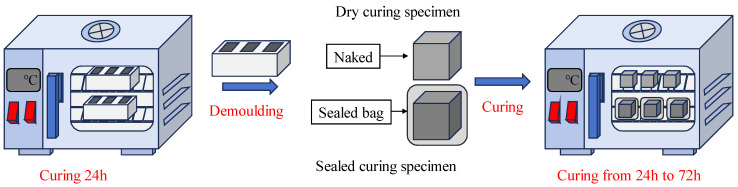
Curing methods and steps.

**Figure 7 materials-17-01413-f007:**
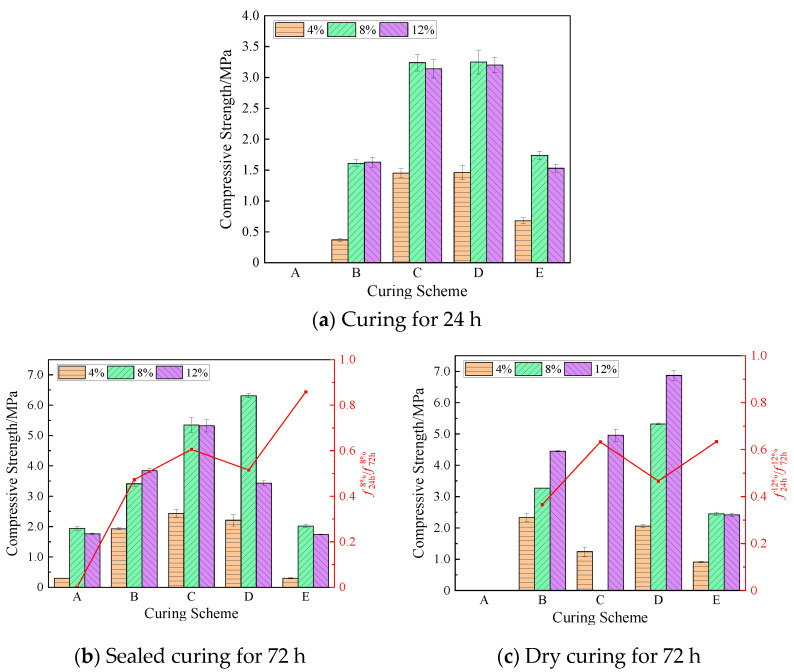
Geopolymer compressive strengths under different temperature curing schemes and curing methods.

**Figure 8 materials-17-01413-f008:**
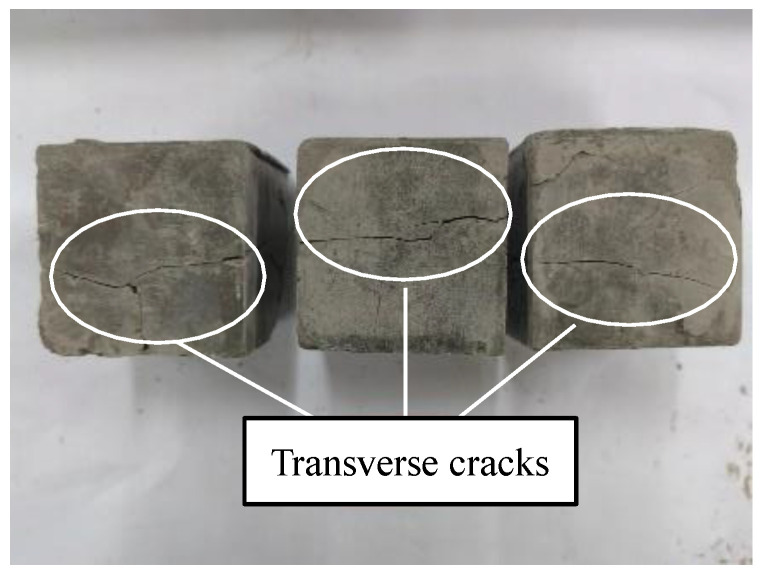
Morphology of simulated lunar soil geopolymer containing 8% NaOH after dry curing for 72 h under scheme C.

**Figure 9 materials-17-01413-f009:**
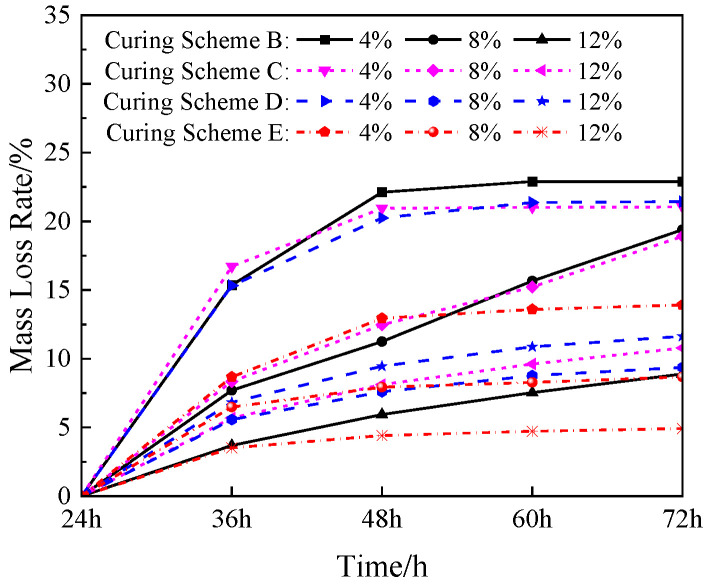
The mass loss rate of geopolymer under dry curing from 24 h to 72 h.

**Figure 10 materials-17-01413-f010:**
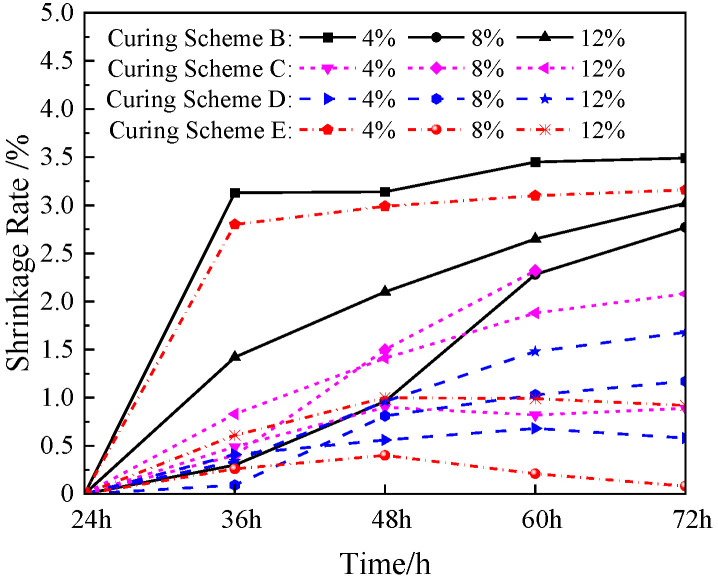
The shrinkage of geopolymer under dry curing from 24 h to 72 h.

**Figure 11 materials-17-01413-f011:**
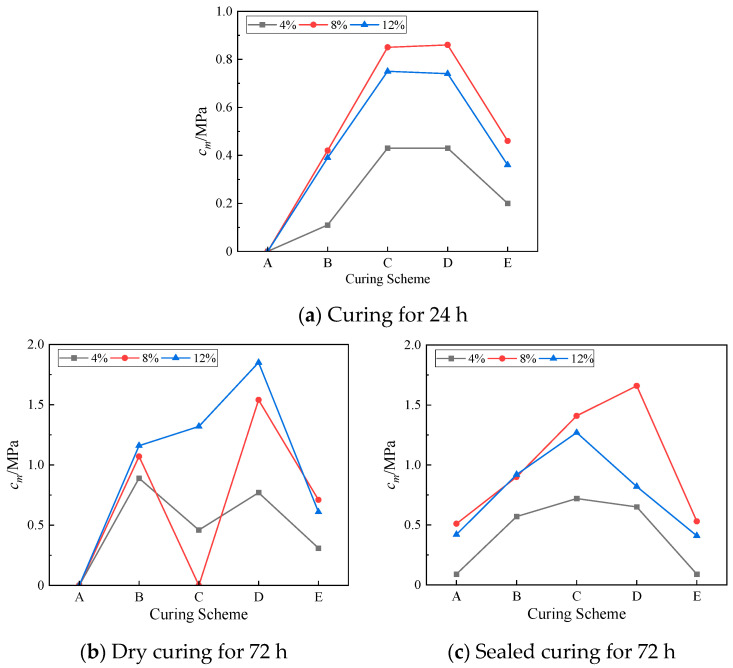
The mass strength efficiency under each curing scheme.

**Figure 12 materials-17-01413-f012:**
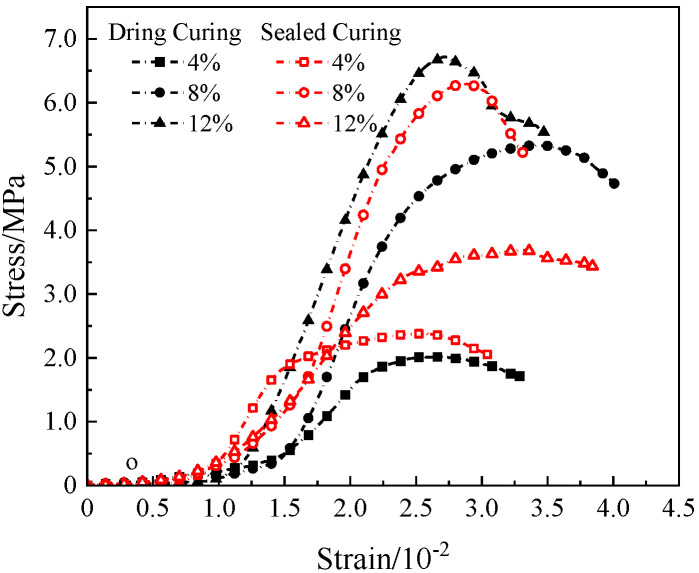
The stress–strain curve of geopolymer in scheme D.

**Figure 13 materials-17-01413-f013:**
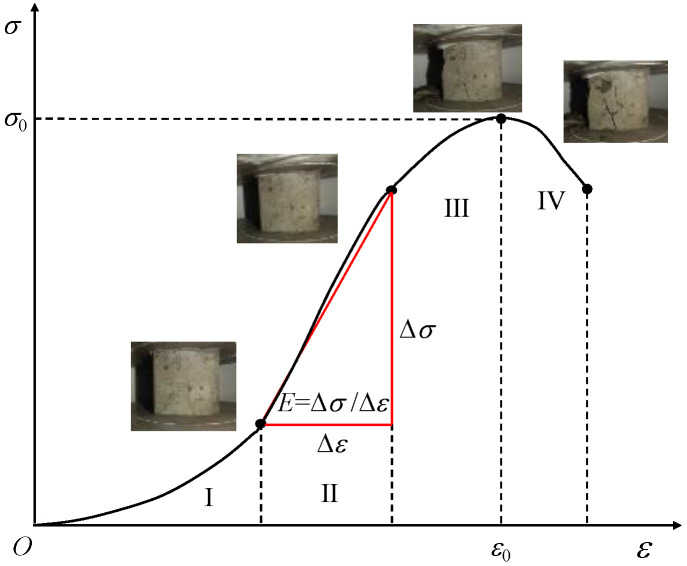
Stress–strain curve and failure mode of lunar soil simulant polymer.

**Figure 14 materials-17-01413-f014:**
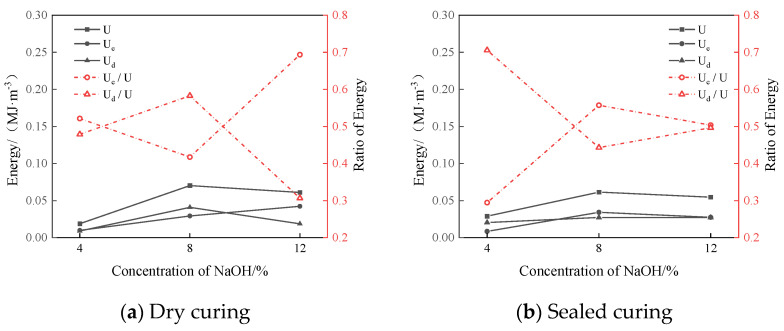
The relationship between polymer energy and NaOH content in lunar soil simulant.

**Figure 15 materials-17-01413-f015:**
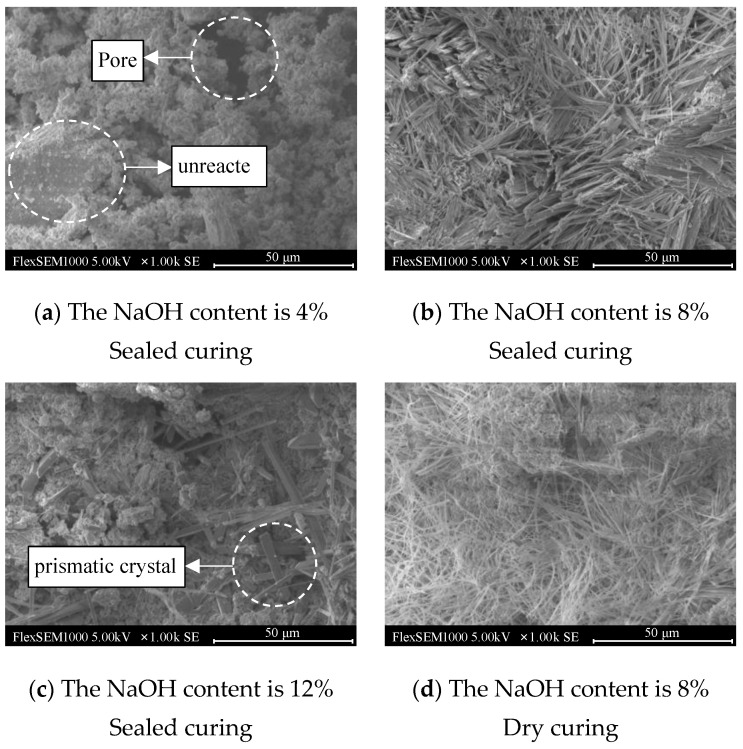
Typical SEM images of geopolymers under D temperature scheme at 72 h.

**Table 1 materials-17-01413-t001:** Compositions of real lunar soil and lunar soil simulant (wt.%).

Item	Apollo-14 [20]	Apollo-15 [20]	JSC-1 [13]	DNA-1 [17]	Experimental Lunar Soil Simulant
SiO_2_	46.30	48.10	47.71	47.79	47.82
Al_2_O_3_	12.90	17.40	15.02	19.16	16.89
FeO	15.10	10.40	10.79	8.75	10.20
CaO	10.70	10.70	10.42	8.28	9.36
MgO	9.30	9.40	9.01	1.86	5.46
Na_2_O	0.54	0.70	2.70	4.38	2.55
K_2_O	0.31	0.55	0.82	3.52	2.30
TiO_2_	3.00	1.70	1.59	1.00	0.06
MnO	0.22	0.14	0.18	-	-

**Table 2 materials-17-01413-t002:** Curing temperature schemes from A to E.

Curing Time	Group of Curing Temperature Scheme				
	A (°C)	B (°C)	C (°C)	D (°C)	E (°C)
0–12 h	5.9	52.7	74.9	76.3	60.3
12–24 h	21.8	60.2	76.2	74.9	52.7
24–36 h	33.5	66.1	77.5	70.6	44.9
36–48 h	45.1	70.5	77.4	66.3	33.7
48–60 h	52.7	74.9	76.3	60.3	22.5
60–72 h	60.2	76.2	74.9	52.7	7.4

**Table 3 materials-17-01413-t003:** Mix proportions of lunar soil simulant geopolymer with different sodium hydroxide contents.

Content of NaOH	*m*_NaOH_/*m*_lss_	*m*_w_/*m*_lss_
4%	0.04	0.3
8%	0.08	0.3
12%	0.12	0.3

*m*_NaOH_ represents the mass of sodium hydroxide particles; *m*_lss_ represents the mass of lunar soil simulant; *m*_w_ represents the mass of mixed distilled water.

## Data Availability

Data are contained within the article.

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
