# Peer review of "Experimental Study on Geopolymerization of Lunar Soil Simulant under Dry Curing and Sealed Curing"

_materials, 2024, doi:10.3390/ma17061413_

Round 1
Reviewer 1 Report
Comments and Suggestions for Authors
The manuscript entitled “Experimental Study on Polymer Solidification of Lunar Soil Simulant Under Dry Curing and Sealed Curing” is in line with the Materials journal. However, this article is based on original research, a lot of information given in this manuscript is not clear and requires to be revised. Because of that, I suggest rejection and encouraging to resubmission of this article to give the authors time to re-write several parts and implement the changes. The areas that have to be corrected are as follows:
· Title: “geopolymerization” instead of “polymerization” should be applied.
· Abstract: The range of temperatures is incorrect.
· Introduction: the information about the planned application “road construction” required to be described more relevant way. What kind of object this way should join? Is a real problem of the speed of the Moon vehicles' lack of road (what about other limitations, including the purpose of usage)? The justification for the provided research is not coherent. This part should be rewritten.
· Introduction: the articles from 2023 connected with the possibility of using geopolymers in extraterrestrial environments are completely ignored.
· Introduction: “Based on above research…” – it is not clear what is common with previously described articles.
· Introduction: Lack of clear definition of research gap and novelty aspects.
· 2.1. Materials – the provided analysis is not sufficient. Material required additional research and comparison with samples of regolith from the Moon (data are available in the literature), especially since there is a lack of comparison of chemical and mineralogical composition as well as morphology. At this moment the claim that the used material is similar for Lunar regolith is not justified.
· Part 2.3. Specimens… the description is not clear additional shame or table with prepared compositions is necessary. There is a lack of explanation for some points of the process. Is possible to wrap the material on the Moon in PE in the first phase of production? Was there any simulation about dry conditions? What kind of bag was used for sealing the samples? Why do the authors use a percentage instead of morality? This part should be rewritten.
· Part 2.4. Test… - there is a lack of description for most methods used in the article, for example, presented in chapters 3.2, 3.3. and others. The description of the research test in this part is too generic. There is for example lack of information about SEM devices and the process of sample reparation for this observation. This part should be rewritten.
· Part 3. 4.: This part should be rewritten, especially the used vocabulary. The content is not connected with quality but rather some aspect of quantity…
· Use small and capital letters in chemical formulas.
· Discussion: A lack of proper discussion part and comparison with up-to-date literature.
· Conclusions: some conclusions are not justified by the provided research, for example, they are connected with the cost of transportation that is not analyzed in the research part or with the appearance of “calcium vanadium…” when there is a lack of chemical analysis in the article.
Reviewer 2 Report
Comments and Suggestions for Authors
The authors report interesting work on realising geopolymers from lunar soil simulants at different NaOH concentrations and temperatures.
There are, in my opinion, numerous shortcomings-errors to be corrected:
- Improve the Introduction by adding more information on geopolymer materials and related literature references.
- In the paragraph describing the materials, the authors report only a graph comparing the chemical composition of the simulated lunar soil
materials with the real lunar soil samples. First, I think there is an error in the graph showing N2O instead of Na2O. Also, I recommend adding a table that at least shows the chemical composition of the simulated lunar soil sample.
- Correct Table 1.
- Insert a table showing the mix design, i.e. the chemical compositions of the reagents/raw materials used to prepare the samples.
- In the microstructural characterisation with SEM micrographs, the authors state (lines 357-358 and the same information is also included in the Conclusions): "There are a large number of gel substances and a small amount of calcium vanadium stone on the surface of the geopolymer".
How come the authors make this statement? Where does the vanadium come from if no vanadium is present in the chemical composition of the simulated lunar soil material and no EDS analysis is reported? I kindly ask the authors to explain and correct this.
- Why did the authors investigate the mechanical properties of materials made at 24 and 72 hours? On the best specimens could have carried out the mechanical tests at 28 days or even 90, as we know that the best mechanical performance of geopolymer materials is recorded at these times.
Comments on the Quality of English Language
In my opinion, the authors should do extensive editing and improvement of the English language.
Reviewer 3 Report
Comments and Suggestions for Authors
Article
Experimental Study on Polymer Solidification of Lunar Soil Simulant Under Dry Curing and Sealed Curing
A very forward-looking article in connection with the inevitable exploitation of raw material deposits on the Moon and the need to transport them on its surface before sending them to Earth. The article provides experimental basis for the application of simulated lunar soil geopolymer in lunar pavement construction.
Comments:
1. the maximum strength of 16 72 h is 6.31 MPa
The maximum strength after 72 hours is 6.31 MPa
2. used JSC-1 to prepare sulfur-rich geopolymer. Although the prob-42
although this can later be deduced from the further content,
it should be explained e.g. JSC-1 new lunar soil simulant
3. According to the 92 method of Reference [16], since the time of a day on the moon is equal to 30 days on the 93
According to the method of [16],
4. the mass ratios of NaOH solid to simulated lunar soil material were 4 %, 8 % and 12 %, 109
is it about the percentage of NaOH addition? because the ratio should be unitless
5. No explanation of what is on the OY axis – „f 8% 24h/f 8% 72h”
6. Table 2. Temperature curing scheme from A to E. 104Curing time, Curing temperature scheme NO.
No- if a number, but there are letters A, B, C...
7. formula is: 189 …m 24h
In the formula: qm represents the mass loss rate of the simulated lunar soil specimen, 190 ∆m represents the difference between the mass of the specimen at each measurement time 191 and the mass of the specimen at 24 h; m24h represents the mass of the specimen at 24 h
Mass loss rate(ML)definition - this definition is very twisted. Mass loss rate from an elementary school textbook: To calculate weight loss percentage, divide the amount of weight lost by your starting weight m0, then multiply that by 100. To break it down, the formula will be: number of pounds lost/your starting weight x 100.
ML1=100%*(m1-m0/m0) at moment 1, ML24=m24-m0/m0 at the end of the tes-24 h
please calculate whether the results from the authors' formula agree with the suggested formula
8. the same : In the formula: ∆l represents the shrinkage rate of the simulated lunar soil specimen; 22 ∆l represents the difference between the axial length of the specimen at each measurement 221 time and the axial length of the specimen at 24h; l24h represents the axial length of the 222specimen at 24h-
maybe these are patterns used in concrete technology?
In my opinion, the reference point should be the initial length of the sample, just like the initial mass of the sample above.
9. ​ It can be seen from Fig.12 (a) that under the dry curing mode, when the NaOH con- 330tent is 12 %, the elastic energy is the largest, which is 0.0423MJ / m3.
Fig. 12 (a) shows that in the dry curing mode, when the NaOH content is 12%, the elastic energy is the highest and amounts to 0.0423 MJ/m3.
10. in lines 330-335 m3 – “3” superscript
11. There are a large number of gel substances and a small amount 357 of calcium vanadium stone on the surface of the geopolymer, and there are holes on the 358
there was no vanadium in the sample –Fig.1???
Round 2
Reviewer 1 Report
Comments and Suggestions for Authors
The manuscript entitled Experimental Study on Geopolymerization of Lunar Soil Simu- 2 lant Under Dry Curing and Sealed Curing” was significantly improved. However, it still required additional changes, as follows:
· Chapter 2.1. Please add also information about the grain shape, for example: https://www.mdpi.com/1996-1073/15/24/9322
· Chapter 3 --> remane as results
· Lack of figures 13 c and d
Author Response
Dear Editors and Reviewers:
Thank you for your letter and for the reviewers’ comments concerning our manuscript entitled “Experimental study on Geopolymerization solidification of lunar soil simulant under dry curing and sealed curing” (ID: materials-2819018). Those comments are all valuable and very helpful for revising and improving our paper, as well as the important guiding significance to our researches. We have studied comments carefully and have made correction which we hope meet with approval.
Reviewer 1:
General Comments:
The manuscript entitled Experimental Study on Geopolymerization of Lunar Soil Simu- 2 lant Under Dry Curing and Sealed Curing” was significantly improved.
Response: Thank you for your affirmation of the content of the article. We gratefully appreciate for your valuable suggestion. Thank you very much for the reviewers ' evaluation of the article. The questions you raised about the article are the key and difficult points encountered in the research. The following will reply to your questions one by one.
Comments and responses:
1. Chapter 2.1. Please add also information about the grain shape, for example: https://www.mdpi.com/1996-1073/15/24/9322
Response: We gratefully appreciate for your valuable suggestion. The content of the paper you recommend is very substantial, and I have a great harvest after reading it. This is a very excellent article on lunar soil simulant and the concept of future lunar base construction. And we put the article as an important reference into the article. Based on your suggestion, I have added a macro photo of the lunar soil simulant material so that readers can better understand the morphology of the lunar soil simulant. As shown in Figure 1 in the revision
2. Chapter 3 --> remane as results
Response: Thank you so much for your careful check. Chapter 3 The title is modified as results.
3. Lack of figures 13 c and d
Response: Figure 13 c and d has been added on.
We appreciate for Editors/Reviewers’ warm work earnestly, and hope the correction will meet with approval. Once again, thank you very much for your comments and suggestions.
Reviewer 2 Report
Comments and Suggestions for Authors
The authors responded comprehensively to my requests for changes and improved the paper.
I recommend the publication of the manuscript.
Author Response
We gratefully thank the editor and all reviewers for their time spend making constructive remarks and useful suggestions, which has significantly raised the quality of manuscript and has enable us to improve the manuscript.
Reviewer 2:
General Comments:
The authors responded comprehensively to my requests for changes and improved the paper.
I recommend the publication of the manuscript.
Response: We are very grateful to you for your affirmation of our paper. Thank you for your valuable time to review our paper. It is our honor to get your recommendation.